# Paraxial mesoderm organoids model development of human somites

**Christoph Budjan[1,2], Shichen Liu[1,2], Adrian Ranga[3], Senjuti Gayen[1,2], Olivier Pourquié[4,5], Sahand Hormoz[1,2,6]\***

[1]Department of Systems Biology, Harvard Medical School, Boston, United States; [2]Department of Data Science, Dana-Farber Cancer Institute, Boston, United States; [3]Laboratory of Bioengineering and Morphogenesis, Biomechanics Section, Department of Mechanical Engineering, Leuven, Belgium; [4]Department of Genetics, Harvard Medical School, Boston, United States; [5]Department of Pathology, Brigham and Women's Hospital, Boston, United States; [6]Broad Institute of MIT and Harvard, Cambridge, United States

**Abstract** During the development of the vertebrate embryo, segmented structures called somites are periodically formed from the presomitic mesoderm (PSM) and give rise to the vertebral column. While somite formation has been studied in several animal models, it is less clear how well this process is conserved in humans. Recent progress has made it possible to study aspects of human paraxial mesoderm (PM) development such as the human segmentation clock *in vitro* using human pluripotent stem cells (hPSCs); however, somite formation has not been observed in these monolayer cultures. Here, we describe the generation of human PM organoids from hPSCs (termed Somitoids), which recapitulate the molecular, morphological, and functional features of PM development, including formation of somite-like structures *in vitro*. Using a quantitative image-based screen, we identify critical parameters such as initial cell number and signaling modulations that reproducibly yielded formation of somite-like structures in our organoid system. In addition, using single-cell RNA-sequencing and 3D imaging, we show that PM organoids both transcriptionally and morphologically resemble their *in vivo* counterparts and can be differentiated into somite derivatives. Our organoid system is reproducible and scalable, allowing for the systematic and quantitative analysis of human spine development and disease *in vitro*.

**\*For correspondence:**
sahand_hormoz@hms.harvard.edu

**Competing interest:** The authors declare that no competing interests exist.

## Editor's evaluation

Budjan et al. describe an organoid protocol to obtain somite-like structures from human iPSCs. Using defined culture media, the authors describe the formation after 5 days *in vitro* of organoids that express a variety of PSM differentiation markers, such as the segmentation clock gene Hes7 and Pax3, thus recapitulating the time course of expression markers typically observed along PSM and somite early differentiation.

## Introduction

Paraxial mesoderm (PM) development involves the formation of embryonic segments called somites, which are produced sequentially from the presomitic mesoderm (PSM) and arranged periodically along the anterior-posterior (AP) axis of the vertebrate embryo. Somites give rise and contribute to a variety of tissues including skeletal muscle, dermis, cartilage, and bone (*Chal and Pourquié, 2017*). Somite formation is controlled by a conserved molecular oscillator, the segmentation clock (*Dequéant et al., 2006*; *Hubaud and Pourquié, 2014*; *Oates et al., 2012*). Previous efforts have focused on how this

**eLife digest** Humans are part of a group of animals called vertebrates, which are all the animals with backbones. Broadly, all vertebrates have a similar body shape with a head at one end and a left and right side that are similar to each other. Although this is not very obvious in humans, vertebrate bodies are derived from pairs of segments arranged from the head to the tail. Each of these segments or somites originates early in embryonic development. Cells from each somite then divide, grow and specialize to form bones such as the vertebrae of the vertebral column, muscles, skin, and other tissues that make up each segment.

Studying different animals during embryonic development has provided insights into how somites form and grow, but it is technically difficult to do and only provides an approximate model of how somites develop in humans. Being able to make and study somites using human cells in the lab would help scientists learn more about how somite formation in humans is regulated.

Budjan et al. grew human stem cells in the lab as three-dimensional structures called organoids, and used chemical signals similar to the ones produced in the embryo during development to make the cells form somites. Various combinations of signals were tested to find the best way to trigger somite formation. Once the somites formed, Budjan et al. measured them and studied their structure and the genes they used. They found that these lab-grown somites have the same size and structure as natural somites and use many of the same genes.

This new organoid model provides a way to study human somite formation and development in the lab for the first time. This can provide insights into the development and evolution of humans and other animals that could then help scientists understand diseases such as the development of abnormal spinal curvature that affects around 1 in 10,000 newborns.

oscillator controls somite formation using a variety of model systems such as mouse, zebrafish, and chick because of ethical and technical limitations of culturing human embryos. Recently, researchers were able to recapitulate PM development using human and mouse pluripotent stem cells cultured as 2D monolayers (*Chu et al., 2019*; *Diaz-Cuadros et al., 2020*; *Matsuda et al., 2020*). These cells undergo species-characteristic oscillations similar to their *in vivo* counterparts. However, final stages of somite development and vertebra formation were not observed in currently published human cell culture systems (*Palla and Blau, 2020*), suggesting that certain aspects of *in vivo* development are not recapitulated in these 2D systems. We reasoned that a 3D cell culture system may exhibit all the stages of PM development including morphogenetic processes associated with somite formation.

Here, we describe an organoid system derived from human induced pluripotent stem cells (hiPSCs) called Somitoids, which faithfully recapitulates functional, morphological, and molecular features of PM development, including formation of somite-like structures *in vitro*. To identify the culture conditions that reproducibly yielded formation of somite-like structures, we developed a quantitative image-based screening platform for individual organoids. Our screening approach identified the optimal parameter values for the culture conditions such as the initial cell number and the concentration of the chemical modulators. We show using single-cell RNA-sequencing (scRNA-seq), immunofluorescence, and qRT-PCR that Somitoids resemble their *in vivo* counterparts both transcriptionally and morphologically and can be differentiated into somite derivatives such as sclerotome and dermomyotome *in vitro*.

Our Somitoid system is reproducible and scalable, allowing for systematic and quantitative analysis of PM development and somite formation to study human spine development and disease in a dish. Finally, our approach can be used to systematically screen organoid cultures for desired phenotypes and reproducibility.

## Results

Recently, protocols have been developed to differentiate mouse or human pluripotent stem cells (hPSCs) towards PM using a combination of the WNT agonist CHIR and BMP inhibitor LDN (*Chal et al., 2015*; *Diaz-Cuadros et al., 2020*). To adapt the protocol for a 3D model of human somitogenesis, we first optimized the initial conditions of our cultures by generating pluripotent spheroids of defined

cell numbers. hiPSCs were allowed to aggregate for 24 hr as suspension cultures in pluripotency media (mTeSR1) in the presence of ROCK inhibitor and polyvinyl alcohol (PVA) to promote aggregation (*Figure 1A*). These pluripotent spheroids resemble cavity-stage epiblast embryos as previously described for kidney organoid cultures (*Freedman et al., 2015*; *Figure 1B*, *Figure 1—figure supplement 1A*). Next, spheroids were cultured in media containing CHIR and LDN (CL), similarly as done for the monolayer cultures (*Chal et al., 2015*; *Diaz-Cuadros et al., 2020*), but with CHIR at a higher concentration (10 µM). After 24 hr in CL media, epiblast-stage cells transition to a neuromesodermal progenitor (NMp) or primitive streak cell fate, characterized by co-expression of T/BRA and SOX2 (*Tzouanacou et al., 2009*; *Figure 1B*). By 48 hr, cells rapidly downregulate SOX2 and express PSM markers, including TBX6 and MSGN1 (*Figure 1B*). This PSM state persists from day 2 to day 4 and is also characterized by the expression of segmentation clock genes such as HES7 (*Figure 1C*). On day 5, organoids showed expression of marker genes associated with somite fate as characterized by qPCR (*Figure 1C*, *Figure 1—figure supplement 1B*). Taken together, the order of activation of marker genes in the PM organoids followed the expected stages of differentiation observed during PM development.

We observed that following the above protocol resulted in a heterogeneous activation of somite marker genes across cells within the same organoid and across different replicates, as well as a low number of somite-like structures (*Figure 1—figure supplement 2*). To improve reproducibility, we set out to screen for the optimal initial cell number, the concentration of the signaling factors, and the timing of their additions during culture. For our primary screen, we compared organoids with an initial cell number of 500 and 1000 as well as combinations of modulators of candidate signaling pathways that have previously been involved in PM development and somite formation, including FGF, WNT, BMP, and TGF-β (*Aulehla et al., 2008*; *Chal et al., 2015*; *Hubaud and Pourquié, 2014*; *Tonegawa et al., 1997*; *Xi et al., 2017*). We primarily focused on the FGF and WNT signaling pathways since their critical role during somitogenesis is well established both *in vivo* (*Aulehla et al., 2008*; *Aulehla et al., 2003*; *Delfini et al., 2005*; *Dubrulle et al., 2001*; *Dunty et al., 2008*; *Greco et al., 1996*; *Yamaguchi et al., 1999*) and *in vitro* (*Chal et al., 2015*; *Sakurai et al., 2012*; *Tan et al., 2013*). Furthermore, dual inhibition of FGF and WNT signaling has been used with some success to generate PM derivatives *in vitro* (*Loh et al., 2016*; *Matsuda et al., 2020*). Finally, the BMP and TGF-β signaling pathways have been shown *in vitro* and *in vivo* to have a role in human somitogenesis (*Loh et al., 2016*; *Xi et al., 2017*).

PSM-stage organoids on day 3 were treated with signaling modulators for 24 and 48 hr, and somite fate and morphogenesis was assessed using PAX3, a somite fate marker, and F-ACTIN, a structural marker of somite formation (*Figure 2A*). We chose day 3 as a starting point for our systematic screen because PSM marker gene expression was more uniform compared to day 2 organoids (*Figure 1B*), and day 4 organoids are not significantly different from day 3 organoids based on a previous study and our own immunostaining and qPCR data (*Figure 1B and C*; *Diaz-Cuadros et al., 2020*). To quantitatively compare conditions, we developed an image analysis pipeline to determine organoid diameter and normalized average PAX3 expression intensity per organoid in an automated manner (*Figure 2C and D*, *Figure 2—figure supplement 2B*). We analyzed three organoids per condition. Strikingly, all organoids that were treated with any combination of FGF or WNT inhibitor reproducibly expressed PAX3 within 24 hr of treatment (*Figure 2B and D*). However, organoids that were initiated from 1000 cells displayed a higher fraction of PAX3-negative cells compared with organoids initiated from 500 cells, even though the average PAX3 expression levels across the entire organoid were comparable (*Figure 2D*, *Figure 2—figure supplement 2A*). Additional staining for SOX2, a neural marker, showed that PAX3-negative cells expressed SOX2, suggesting that our PM organoids derive from NMps (*Figure 1B*, *Figure 2—figure supplement 2A*). In addition, staining for F-ACTIN, together with nuclear expression of PAX3, more consistently revealed somite-like structures (radial arrangement of PAX3+ columnar cells with expression of apical F-ACTIN in the central cavity) in organoids made from 500 cells compared with organoids made from 1000 cells (*Figure 2B*, *Figure 2—figure supplement 3A and B*). Taken together, based on these observations, we used 500 cells as the initial cell number going forward.

Organoids that were treated for 48 hr with signaling pathway modulators showed an overall decrease of PAX3 expression compared with organoids treated for only 24 hr across replicates, indicating that prolonged signaling manipulation does not improve the somite phenotype (*Figure 2D*).

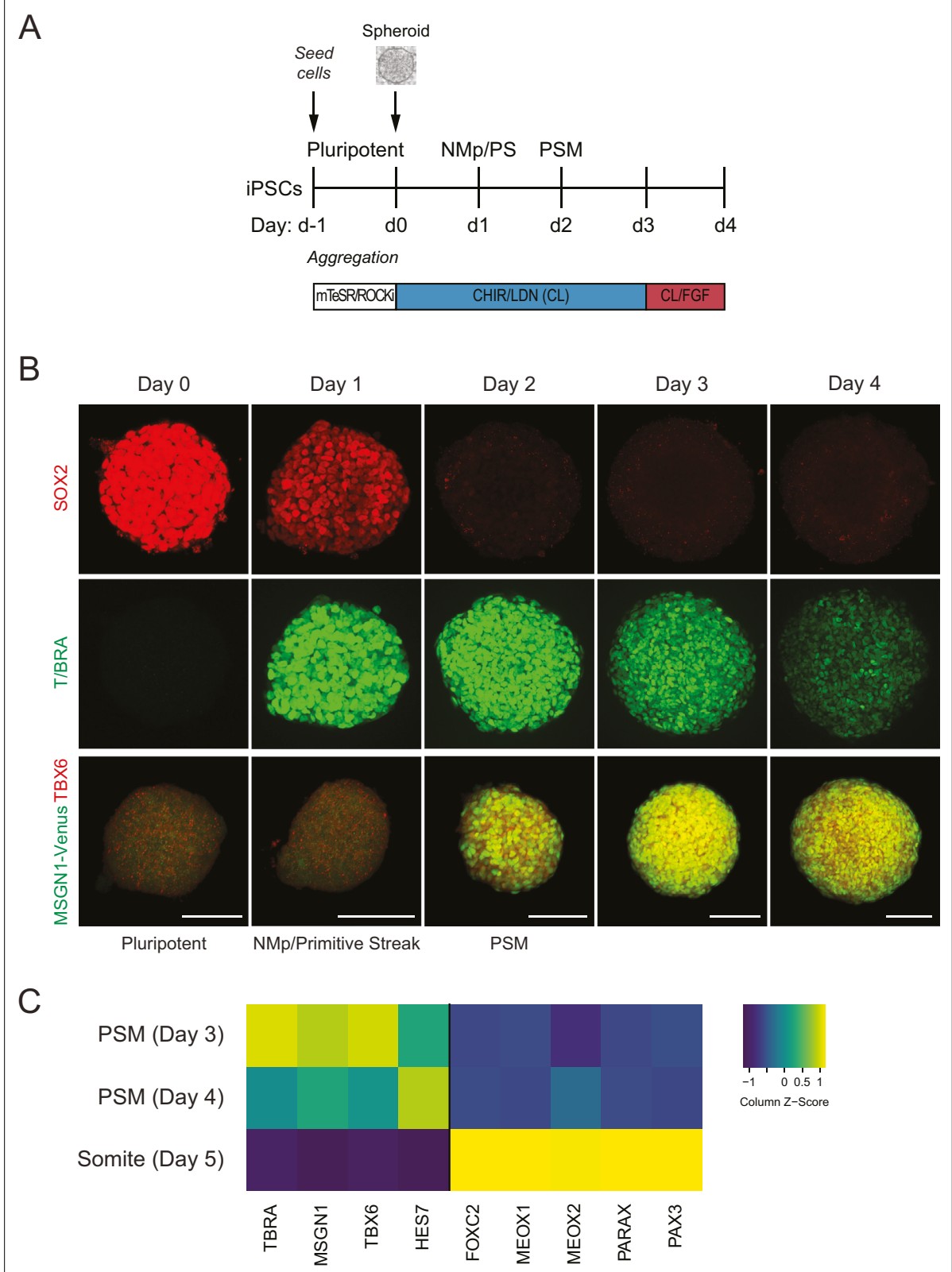

**Figure 1.** Human pluripotent stem cell (hPSC)-derived paraxial mesoderm (PM) organoids turn on marker genes associated with PM differentiation. (**A**) Schematic overview of PM organoid differentiation protocol from hPSCs. hPSCs aggregated and formed spheroids for 24 hr prior to differentiation. For differentiation, spheroids were exposed to Wnt agonist (CHIR) and BMP inhibitor (LDN) for 72 hr. On day 3, FGF2 was added to the media in addition to CHIR and LDN. (**B**) Immunofluorescence analysis of cell fate-specific marker genes shows progressive differentiation towards presomitic

*Figure 1 continued on next page*

*Figure 1 continued*

mesoderm (PSM) fate (top and middle rows). Organoids derived from human induced pluripotent stem cells (hiPSCs) harboring an MSGN1-Venus reporter express TBX6 at the same time as the reporter is activated (bottom row). Scale bar represents 100 μm. Representative images shown from n = 3 independent experiments. Cell lines used: NCRM1 hiPSCs and MSGN1-Venus hiPS reporter cells. (**C**) qRT-PCR analysis of PSM and somite markers reveals PSM-to-somite transition from day 4 to day 5. Relative gene expression levels are shown as Z-scores, expressed as fold-change relative to undifferentiated iPSCs (see Materials and methods). Source data is available in *Figure 1—source data 1*.

The online version of this article includes the following source data and figure supplement(s) for figure 1:

**Source data 1.** qPCR raw data of paraxial mesoderm (PM) organoid differentiation.

**Figure supplement 1.** Additional immunofluorescent and qPCR data.

**Figure supplement 2.** Organoids generated using an unoptimized protocol exhibit heterogeneous activation of somite marker genes (PAX3 and NCAD) and a low number of rosette structures.

Additionally, organoids initiated from 500 cells and then treated for 48 hr had a smaller diameter compared with organoids of the same initial cell number that were treated for only 24 hr (*Figure 2C*). This suggests that long-term inhibition of WNT and/or FGF, known mitogenic signaling pathways, has detrimental effects on proliferation or cell survival. These results indicate that treatment of PSM-stage organoids with pathway modulators for 24 hr is sufficient to induce somite fate.

Next, we set out to optimize the culture conditions to increase the number of somite-like structures in addition to the expression levels of the somite marker genes. We looked for morphological hallmarks of somite formation, specifically the formation of rosette-like structures consisting of radially arranged bottle-shaped PAX3+ epithelial cells with their NCAD+ apical surface facing a central cavity (*Figure 3—figure supplement 1A and B*, *Figure 3—video 1*).

We used 500-cell spheroids as an initial starting point and compared two different inhibitor doses for FGF and WNT in addition to the other pathway modulators applied to PSM-stage organoids on day 3. We characterized the organoids after treating them for 24 hr, 48 hr, and 24 hr followed by culture in basal media without any added factor for an additional 24 hr (five organoids per condition, *Figure 3A*). In addition to quantifying PAX3 levels (*Figure 3D*, *Figure 3—figure supplement 2B*), we also counted the number of somite-like structures per organoid (*Figure 3B and E*, *Figure 3—figure supplement 2A and B*; see Materials and methods for a description of the scoring criteria). Comparing PAX3 expression levels in our treated organoids, we observed that somite fate can be broadly induced across a range of treatment regimes, concentrations, and types of WNT and/or FGF inhibitors (*Figure 3C and D*, *Figure 3—figure supplement 2C*). However, the number of somite-like structures is not necessarily correlated with average PAX3 expression levels. For example, the numbers of somite-like structures in several conditions (FGFRi$^{hi}$/PD173, WNTi$^{hi}$/C59, WNT$^{hi}$/C59+ FGFRi$^{lo}$, WNTi$^{hi}$/XAV) were lower in organoids treated for 24 hr followed by culture in basal media for 24 hr compared with organoids that were treated with the same inhibitors for 48 hr, even though they exhibited higher PAX3 expression levels on average (*Figure 3D and E*, *Figure 3—figure supplement 2A*). This suggests that marker gene expression alone may not be the best predictor when screening for morphologically complex phenotypes such as somite formation.

Surprisingly, organoids that were cultured for an additional 24 hr (day 3 to day 4) in FGF, WNT pathway agonist, and BMP inhibitor, considered a treatment control, followed by culture in basal media only for another 24 hr, consistently exhibited the highest number of somite-like structures across all organoid replicates (*Figure 3E*, *Figure 3—figure supplement 1A and B*, *Figure 3—figure supplement 2A and B*, *Figure 3—video 1*) as well as technical replicates (*Figure 3—figure supplement 3A*). Additionally, the average PAX3 expression was among the highest of all conditions tested (*Figure 3D*). This suggests that simply removing FGF/WNT pathway agonists and BMP inhibitor, which maintain cells in a PSM state, is sufficient to reproducibly induce somite fate and morphological formation of somite-like structures (*Figure 3—figure supplement 2A*, *Figure 3—figure supplement 3A*). Computing the variation of the number of somite-like structures across the five organoids confirmed that this phenotype was highly reproducible (coefficient of variation = 11.1%; *Figure 3—figure supplement 2B*). Finally, our optimized protocol reproducibly yielded efficient induction of somite-like structures in multiple genetically independent hiPSC lines (*Figure 3—figure supplement 3B*). Taken together, we determined that initiating the protocol with 500 cells and treating day 3 organoids with CL + FGF for 24 hr followed by culture in basal media for an additional 24 hr yield the most robust induction of somite-like structures while minimizing variation between experiments (technical

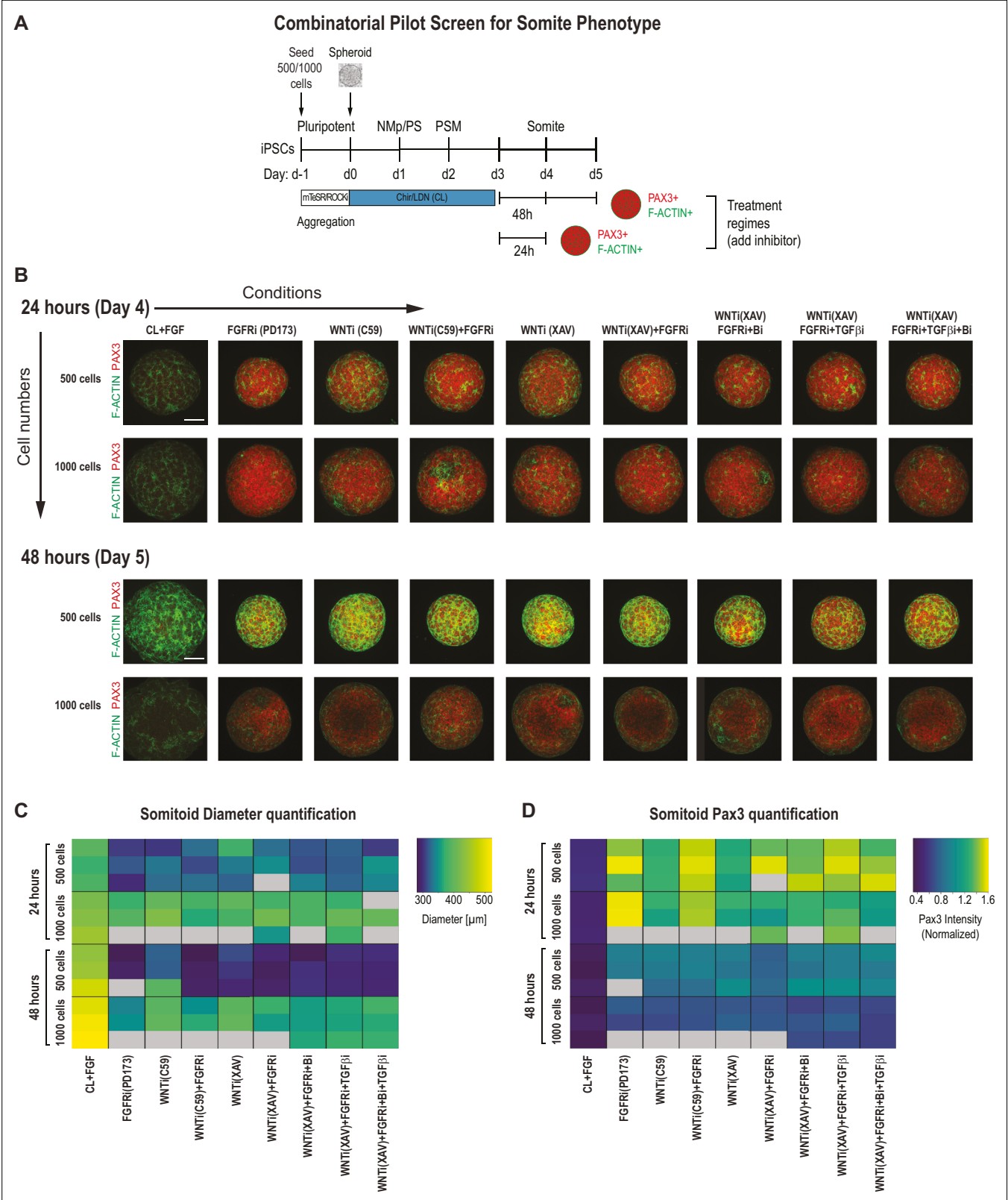

**Figure 2.** Pilot screen to optimize differentiation conditions for somite phenotype in paraxial mesoderm (PM) organoids reveals optimal initial number of cells and duration of treatment. (**A**) Schematic overview of systematic screen in PM organoids (Somitoids). Presomitic mesoderm (PSM)-stage organoids were treated on day 3 for 24 hr or 48 hr with signaling agonists/antagonists. Treated organoids were cultured in basal media with inhibitors as indicated. Control organoids were maintained in CL media with FGF added. NCRM1-derived organoids were used for the screen. (**B**) Representative

*Figure 2 continued on next page*

*Figure 2 continued*

immunofluorescent images of day 4 and day 5 organoids after treatment for 24 hr or 48 hr, respectively, stained for somite marker PAX3 and F-ACTIN to visualize rosette-like somite structures. Organoids generated from 1000 cells generally show a more diffuse F-ACTIN pattern compared to organoids made from 500 cells, which exhibit bright foci, consistent with somite formation. Confocal images are shown as maximum intensity z-projections. Scale bar represents 100 μm. Small-molecule inhibitors used are indicated in brackets. FGFRi, FGF receptor inhibitor (PD173074); WNTi; Wnt inhibitor (C59 or XAV939); Bi, BMP inhibitor (LDN); TGF-βi, TGF-β inhibitor (A-83-01). Representative image shown for each condition from three organoid replicates. (**C**) Automated quantification of organoid diameter for each organoid/replicate treated as indicated (see Materials and methods for details). Three organoids per condition were characterized except where indicated with gray boxes. Organoids initiated from 500 cells show a decreased diameter when treated for 48 hr compared with 24 hr. Source data is available in *Figure 2—source data 1*. (**D**) Automated quantification of normalized average PAX3 intensity for each organoid/replicate treated as indicated. Three organoids are shown per condition except where indicated with gray boxes. Organoids initiated from both 500 and 1000 cells show higher average normalized PAX3 levels when treated for 24 hr compared with 48 hr. Source data is available in *Figure 2—source data 2*.

The online version of this article includes the following source data and figure supplement(s) for figure 2:

**Source data 1.** Quantification of organoid diameter from primary screen.

**Source data 2.** Quantification of average PAX3 levels per organoid from primary screen.

**Figure supplement 1.** Replicate data of pilot screen for somite phenotype in human paraxial mesoderm (PM) organoids.

**Figure supplement 2.** Additional immunofluorescent data and inter-organoid phenotypic variance of primary somite phenotype screen in Somitoids.

**Figure supplement 3.** Organoids made from 500 cells more reproducibly generate somite-like structures compared to organoids made from 1000 cells.

variation) and different cell lines (biological variation). This optimized differentiation protocol was therefore used for all subsequent experiments.

To further characterize the developmental trajectory and transcriptional states of our Somitoid system, we collected 15,558 cells (after postprocessing) over the course of the optimized 5-day differentiation protocol at timepoints that capture the key transition steps (day 1, day 2, day 3, day 5) and performed scRNA-seq (*Figure 4A*). Multiple organoids were used to obtain the required number of cells at each timepoint (see Materials and methods). We first combined all the cells across the timepoints and clustered them using the Leiden clustering algorithm (*Traag et al., 2019*). Predominantly, the four major clusters corresponded to cells from the four different timepoints. Therefore, cells at each timepoint have transcriptional states that are different compared with cells from the other timepoints. In addition, within each timepoint, cells exhibit similar transcriptional profiles as indicated by the uniformity of the expression levels of marker genes across individual cells (*Figure 4E*, *Figure 4— figure supplements 1–3*).

Cells collected on day 1 exhibited gene expression profiles similar to primitive streak or NMps, expressing genes such as SOX2, T/BRA, MIXL1, and NODAL (*Figure 4C and E*, *Figure 4—figure supplement 1*). Starting on day 2, cells expressed canonical PSM marker genes such as TBX6, MSGN1, WNT3A, RSPO3, and clock genes of the Notch signaling pathway including HES7, LFNG, DLL1, and DLL3 (*Figure 4C and E*, *Figure 4—figure supplement 2*). Day 5 cells expressed somite marker genes such as TCF15, PAX3, FOXC2, and MEOX2 (*Figure 4C and E*, *Figure 4—figure supplement 3*). Interestingly, a subset of the day 5 cells also expressed somite polarity markers, UNCX and TBX18, which suggests faithful recapitulation of somite patterning in Somitoids (*Figure 4C*, *Figure 4—figure supplement 3*). Furthermore, two of the subclusters ('PSM-to-somite,' 'early somite'), which are characterized by co-expression of PSM and somite marker genes, comprised both day 3 and day 5 cells, indicating that the PSM-to-somite transition is captured in our *in vitro* system (*Figure 4A and B*). Interestingly, one somite subcluster ('late somite') was enriched for myogenic genes (MYL4/6/7/9, TROPONIN L1) and sclerotome genes (TWIST1, COL1A1, COL11A1, COL7A1, ACTA2), suggesting that these cells represent more downstream fates of somite-derived cells (*Figure 4B*, *Figure 4— figure supplement 4A*). Finally, we observed the expected sequential activation pattern of the HOX genes in our Somitoid system starting with HOXA1 on day 1, followed by other cervical and thoracic HOX genes on days 2–3, to HOXD9, a lumbosacral HOX gene, in the somite-stage organoids (day 5; *Figure 4D*, *Figure 4—figure supplement 4B*). Taken together, our scRNA-seq analyses show that our Somitoid system faithfully recapitulates the gene regulatory programs of human PM development and generates mature somite-like cells, which express the full repertoire of known marker genes. Furthermore, we did not detect cells of different developmental origins, suggesting that we are generating homogeneous organoids containing only PM derivatives.

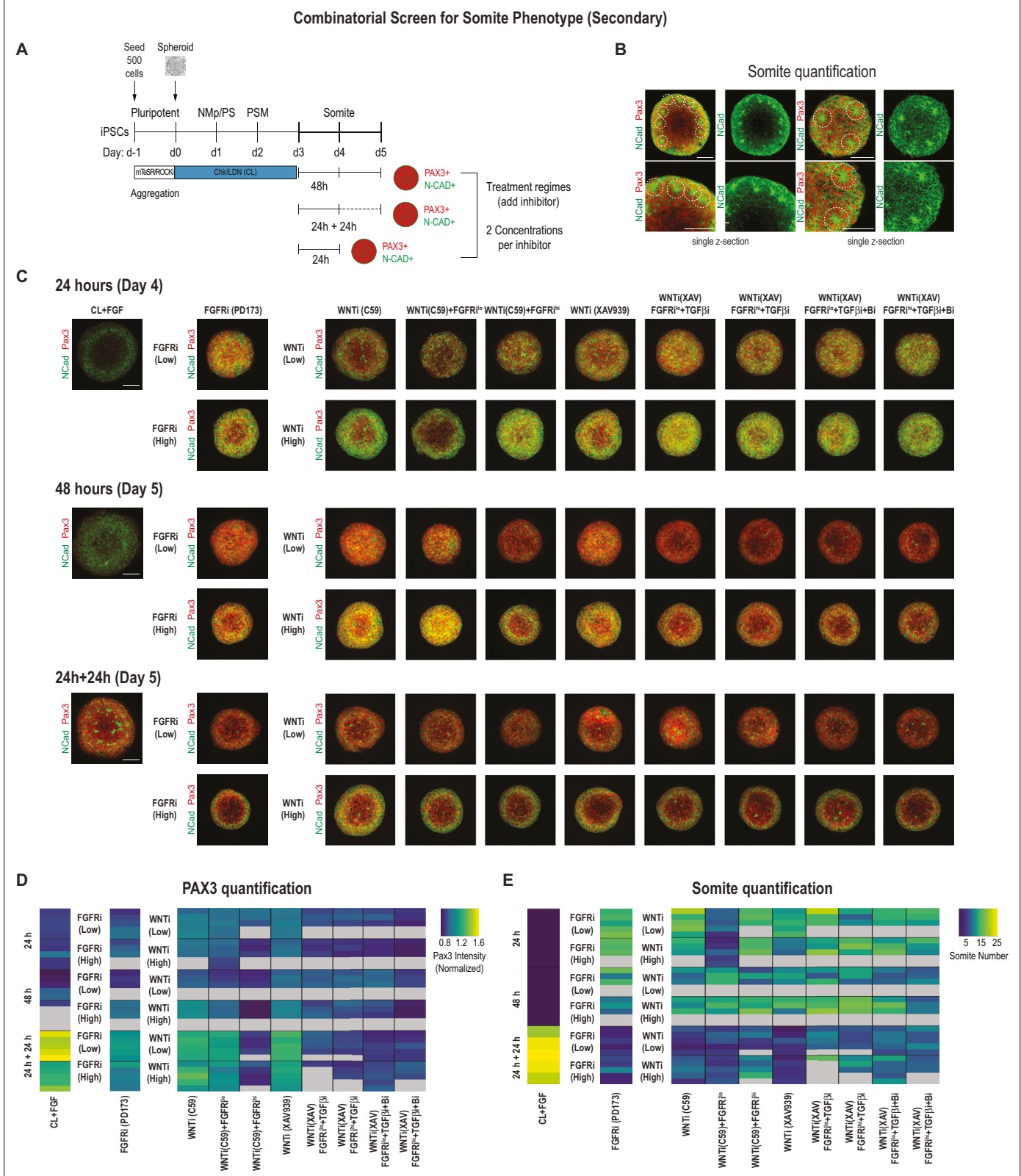

**Figure 3.** Secondary screen of paraxial mesoderm (PM) organoids identifies optimal differentiation protocol for somite formation. (**A**) Schematic overview of secondary screen in PM organoids. Presomitic mesoderm (PSM)-stage organoids were treated on day 3 for 24 hr followed by measurement, 24 hr of treatment followed by 24 hr culture in basal media (no added factors) and then measured (24 hr + 24 hr), and 48 hr of treatment followed by measurement. Treated organoids were cultured in basal media with inhibitors as indicated. WNT and FGF inhibitors were tested at two

*Figure 3 continued on next page*

*Figure 3 continued*

different concentrations. Control organoids were maintained in CL media with FGF added. NCRM1-derived organoids were used for the screen. (**B**) Representative immunofluorescent images of day 5 organoids stained for somite markers PAX3 and NCAD showing the rosette-like structures that were scored as somite-like structures based on expression of somite fate markers and structural features (scoring criteria detailed in Materials and methods). Images are shown as individual z-sections. (**C**) Representative immunofluorescent images of day 4 and day 5 organoids stained for somite markers PAX3 and NCAD to visualize rosette-like somite structures. Confocal images are shown as maximum intensity z-projections. Scale bar represents 100 μm. Small-molecule inhibitors used are indicated in brackets. FGFRi, FGF receptor inhibitor (PD173074); WNTi, Wnt inhibitor (C59 or XAV939); Bi, BMP inhibitor (LDN); TGF-βi, TGF-β inhibitor (A-83-01). Representative image shown for each condition from five organoid replicates. (**D**) Automated quantification of normalized average PAX3 intensity for each organoid/replicate treated as indicated (see Materials and methods for details). Five organoids are shown per condition except as indicated with gray boxes. Several inhibitor combinations with a treatment regime of 24 hr treatment followed by 24 hr cultured in basal media show highest average PAX3 levels. Source data is available in *Figure 3—source data 1*. (**E**) Quantification of the number of somite-like structures for each condition. Each row represents one organoid replicate. Five organoids are shown per condition except where indicated with gray boxes. Organoids that were maintained in CL media with added FGF for 24 hr followed by culture in basal media for 24 hr reproducibly exhibit the highest number of somite-like structures per organoid. Source data is available in *Figure 3—source data 2*.

The online version of this article includes the following video, source data, and figure supplement(s) for figure 3:

**Source data 1.** Quantification of average PAX3 levels per organoid from secondary screen.

**Source data 2.** Quantification of somite-like structures of secondary screen.

**Figure supplement 1.** High-resolution imaging of somite-like structures in day 5 organoids.

**Figure supplement 2.** Additional qPCR data and inter-organoid phenotypic variance of secondary somite phenotype screen in Somitoids.

**Figure supplement 2—source data 1.** qPCR data of calculated expression levels of selected treatment regimes from secondary screen.

**Figure supplement 3.** The optimized Somitoid protocol is reproducible across experiments and different cell lines.

**Figure supplement 3—source data 1.** Quantification of somite-like structures of technical replicates using the NCRM1 cell line.

**Figure supplement 3—source data 2.** Quantification of somite-like structures of biological replicates using the NCRM1, ACTB-GFP, and WTC cell lines.

**Figure 3—video 1.** Confocal z-stacks of Somitoids and control organoids immunostained for PAX3 and NCAD showing *in vitro* somite-like structures.
https://elifesciences.org/articles/68925/figures#fig3video1

We next assessed whether our *in vitro*-derived somite-like structures show similar spatial organization and size distribution to their *in vivo* counterparts. To independently confirm some of the somite marker genes that we identified in our scRNA-seq dataset, we measured expression levels of TCF15/PARAXIS, PAX3, and F-ACTIN in day 5 Somitoids using whole-mount immunostaining (*Figure 5A*). Day 5 cells co-expressed both somite marker genes TCF15 and PAX3 throughout the Somitoid, and somite-like structures displayed apical localization of F-ACTIN. To determine whether *in vitro*-derived somite-like structures were similar in size to human embryonic segments, we compared them with Carnegie stage 9–11 early human somites (*Figure 5B*; see Materials and methods for a description of how somite sizes were quantified). Organoid-derived somite-like structures were similar in size (median area = 8892 μm², interquartile range [IQR] = 7698–10682 μm²) to Carnegie stage 11 somites (median area = 9681 μm², IQR = 8262–11493 μm²) but larger than earlier-stage human somites (Carnegie stage 9 somites, median area = 4399 μm², IQR = 4089–4433 μm²; Carnegie stage 10 somites, median area = 4704 μm², IQR = 4477–5343 μm²; see *Figure 5B*). Together, these results suggest that our organoid-derived somite-like structures share spatial and molecular features as well as overall size with their *in vivo* counterparts.

Finally, we assessed whether Somitoids can give rise to downstream PM derivatives of sclerotome and dermomyotome. First, we differentiated Somitoids to sclerotome by exposing day 5 organoids to SHH agonist and WNT inhibitor to mimic the signaling environment of ventral somites *in vivo* (*Fan et al., 1995*; *Fan and Tessier-Lavigne, 1994*; *Loh et al., 2016*). After 3 days, the organoids showed a robust induction of canonical sclerotome marker genes such as PAX1, SOX9, and COL2A1 (*Figure 5C*). Additionally, day 5 Somitoids were differentiated towards dermomyotome by exposing them to WNT/BMP agonists and SHH inhibitor for 48 hr followed by dissociation and culture on Matrigel as a monolayer in muscle differentiation medium (*Loh et al., 2016*; *Matsuda et al., 2020*) to further differentiate them to skeletal muscle. Immunostaining for myosin heavy chain (MYH1, a myocyte/myotube marker) confirmed that our Somitoid-derived cells can generate skeletal muscle derivatives *in vitro* (*Figure 5—figure supplement 1*). These data demonstrate that our *in vitro*-induced

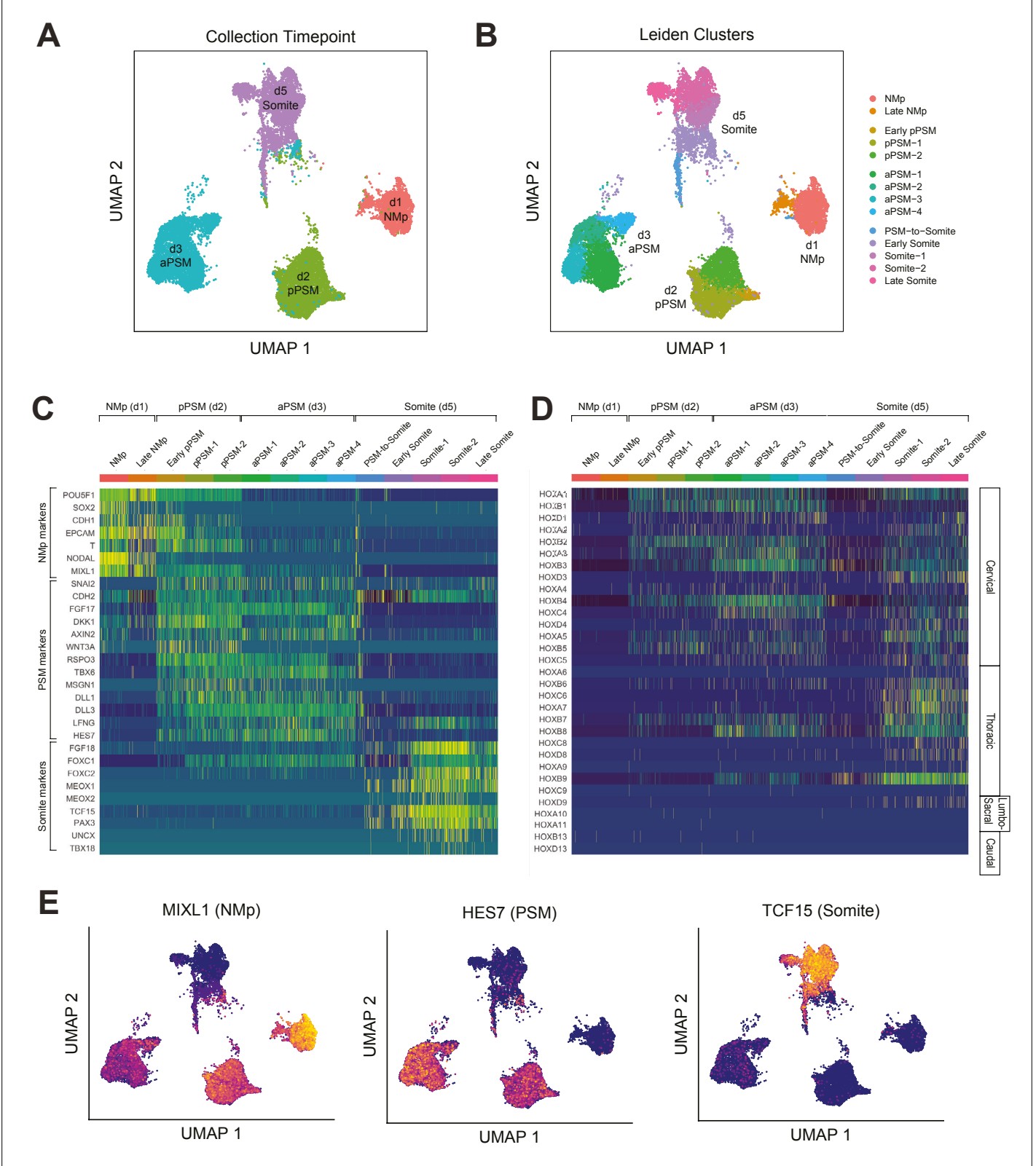

**Figure 4.** Single-cell RNA-sequencing analysis of paraxial mesoderm PM organoids (Somitoids) reveals differentiation trajectory from NMp-like cells to somite-stage PM. (**A**) Uniform Manifold Approximation and Projection (UMAP) of single-cell transcriptomes of differentiating human PM organoids, colored by collection timepoint (15,558 cells). NCRM1-derived organoids were used to collect single cells. (**B**) UMAP of human PM organoids, colored by assigned Leiden cluster identity based on marker gene expression profile (see Materials and methods). (**C**) Heatmap of selected marker genes of

*Figure 4 continued on next page*

*Figure 4 continued*

PM differentiation. Collection timepoint and Leiden cluster identities are indicated. Marker genes are grouped based on primary associated cell fate as indicated. (**D**) Heatmap of single-cell HOX gene expression levels. Cells are grouped by Leiden cluster identity. Hox genes are ordered by position, with anatomical positions of HOX paralogues indicated on the right. (**E**) UMAP plots overlaid with normalized transcript counts of representative cell fate marker genes.

The online version of this article includes the following figure supplement(s) for figure 4:

**Figure supplement 1.** Single-cell RNA-sequencing analysis of differentiating human paraxial mesoderm (PM) organoids, neuromesodermal progenitor (NMp) marker genes.

**Figure supplement 2.** Single-cell RNA-sequencing analysis of differentiating human paraxial mesoderm (PM) organoids, presomitic mesoderm (PSM) marker genes.

**Figure supplement 3.** Single-cell RNA-sequencing analysis of differentiating human paraxial mesoderm (PM) organoids, somite marker genes.

**Figure supplement 4.** Single-cell RNA-sequencing analysis of differentiating human paraxial mesoderm (PM) organoids, cluster-based marker gene identification, and HOX gene analysis.

**Figure supplement 5.** Single-cell RNA-sequencing analysis of differentiating Somitoids reveals downregulation of WNT, FGF, and NOTCH target genes in day 5 somite-like cells.

somite-like structures maintain their ability to differentiate further into somitic mesoderm derivatives of the sclerotome and dermomytome lineages.

## Discussion

Here, we reported the generation of human PM organoids from hPSCs that reproduce important features of somitogenesis not previously captured in conventional monolayer differentiation cultures, most notably formation of somite-like structures. Using a simple suspension culture that does not require manual matrix embedding, we identified optimal differentiation conditions by systematically screening initial cell numbers and modulating the signaling factors. Importantly, our culture conditions are compatible with high-throughput screening approaches. Many established organoid protocols currently have limited applications because they are not reproducible. Therefore, we set out to identify the optimal differentiation conditions that minimized the variability between organoids as quantified by automated image analysis.

One critical parameter we identified in our screens was the initial cell number used for aggregation. Our results suggest that if the initial cell number is above a certain threshold then somite fate cannot be induced in a homogeneous manner in our organoid system. This result is in line with previous findings in 3D models such as gastruloids, multi-axial self-organizing aggregates of mouse ES cells, which exhibit a higher degree of variability and multiple elongations when the number of initial cells exceeds a threshold (*Beccari et al., 2018*; *van den Brink et al., 2014*). Another important finding of our screens was that simply removing FGF, WNT pathway agonist as well as BMP inhibitor yielded the most reproducible and efficient somite-like structure-forming organoids. This treatment regime does not necessarily follow from applying prior *in vivo* and *in vitro* knowledge of somitogenesis. Previous protocols have used FGF and WNT inhibitors (*Matsuda et al., 2020*) or inhibition of all four candidate signaling pathways (FGF, WNT, BMP, and TGF-β) to induce somite fate (*Loh et al., 2016*) in monolayer cultured hiPSCs. While these conditions similarly induced somite fate marker genes in our 3D system, removal of FGF/WNT agonists and BMP inhibitor overall performed better as indicated by larger organoid diameters, higher average PAX3 expression levels, and higher number of somite-like structures. In line with these findings, our scRNA-seq analysis revealed that day 5 somite-like cells from our optimized protocol autonomously downregulate WNT target genes (DKK1, AXIN2, WNT3A, RSPO3; *Figure 4—figure supplement 2*) and FGF target genes (FGF8, FGF17, SPRY4, DUSP6/MKP3, SEF/IL17RD; *Figure 4—figure supplement 5*).

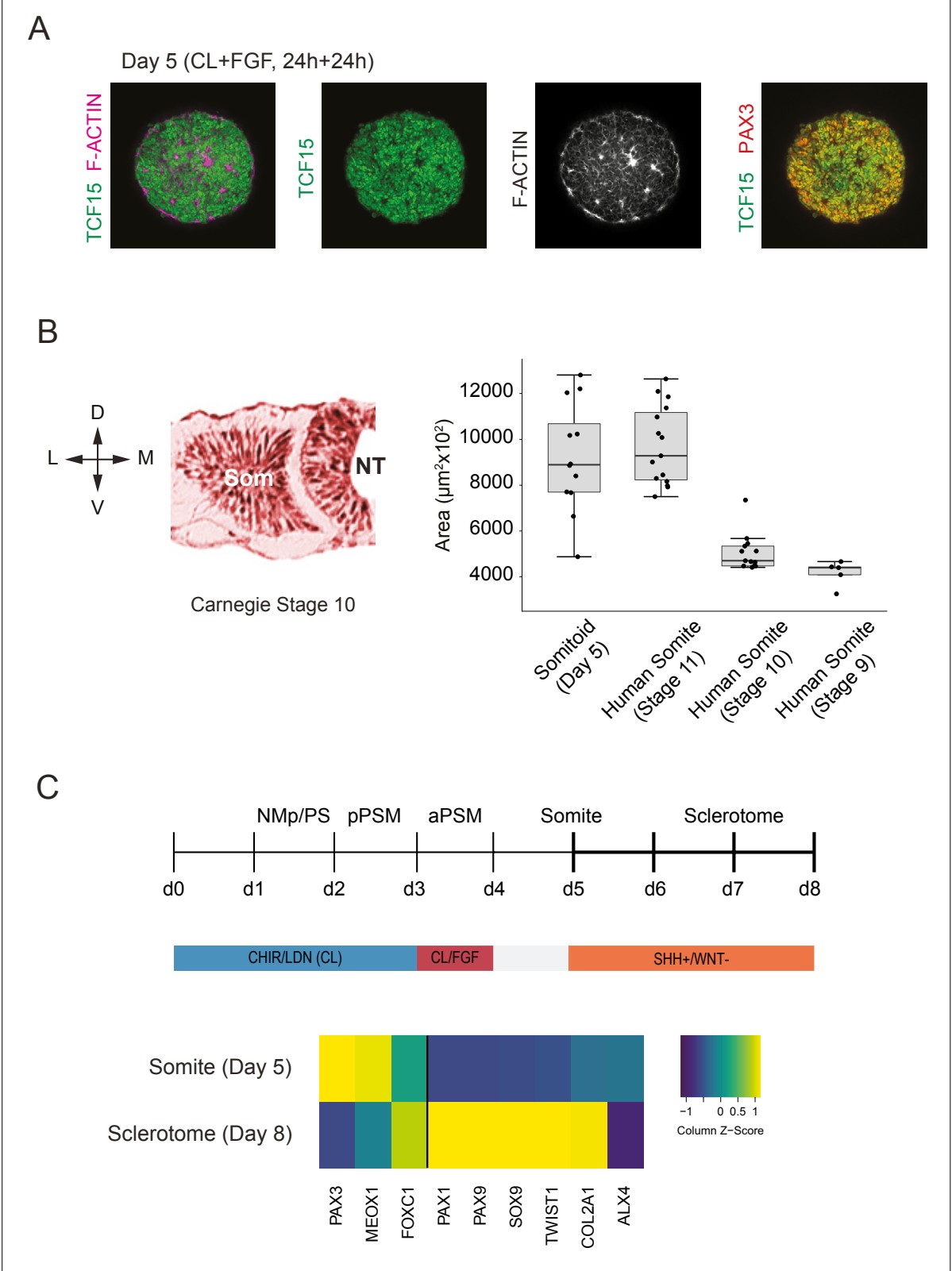

**Figure 5.** Somitoids express known somite-stage-specific marker genes and can differentiate to sclerotome fate. (**A**) Whole-mount Immunofluorescence analysis of day 5 Somitoids reveals co-expression of somite markers TCF15/PARAXIS and PAX3 and polarized rosette-like structures as indicated by F-ACTIN localization, suggesting that somite-like structures resemble *in vivo* counterparts in both a molecular and morphological manner. Representative images are shown as maximum intensity z-projections from three organoid replicates. NCRM1 human induced pluripotent stem cells (hiPSCs) were used

*Figure 5 continued on next page*

*Figure 5 continued*

to generate Somitoids. (**B**) Quantification of somite-like structure sizes in day 5 human Somitoids and human embryos (Carnegie stages 9–11) reveals that the median and interquartile range of *in vitro* somite-like structure sizes (calculated as area) is comparable to Carnegie stage 11 human somites *in vivo* (see Materials and methods). Boxes indicate interquartile range (25th percentile to 75th percentile). End of whiskers indicates minimum and maximum. Points indicate individual somite-like structures. Central lines represent the median. Carnegie embryo data were obtained from the Virtual Human Embryo Project (https://www.ehd.org/virtual-human-embryo). NCRM1 hiPSCs were used to generate Somitoids. Source data is available in *Figure 5—source data 1*. (**C**) Sclerotome differentiation of Somitoids. Day 5 Somitoids were exposed to SHH agonist and WNT inhibitors to induce sclerotome differentiation as indicated. qPCR analysis of somite and sclerotome markers reveals induction of sclerotome markers on day 8. Relative gene expression levels are shown as Z-scores, expressed as fold-change relative to undifferentiated iPSCs (see Materials and methods). NCRM1 hiPSCs were used to generate Somitoids. Source data is available in *Figure 5—source data 2*.

The online version of this article includes the following source data and figure supplement(s) for figure 5:

**Source data 1.** Comparative somite size quantification of *in vitro* somite-like structures and human somites from the Carnegie collection.

**Source data 2.** qPCR raw data of sclerotome differentiation.

**Figure supplement 1.** Differentiation of Somitoid-derived cells towards skeletal muscle.

Finally, our screening results also suggest that focusing on marker gene induction as a phenotypic readout alone is not sufficient to optimize culture conditions of more complex organoid models such as somitogenesis.

The scRNA-seq analysis of our hPSC-derived Somitoids independently confirmed our immunostaining and qRT-PCR results and showed that all major paraxial mesodermal cell types are generated, consistent with the cell types observed during PM development. Comparing our single-cell dataset with previously published *in vitro*-generated human PM transcriptomic data of monolayer cultures (*Diaz-Cuadros et al., 2020*; *Matsuda et al., 2020*) reveals a similar pattern of activation of marker genes. Diaz-Cuadros and colleagues did not generate bona fide somitic cells as their final timepoint cell population does not express canonical somitic markers. Matsuda et al. indeed show expression of several somitic marker genes including TCF15, MEOX1, and PAX3 based on bulk RNA-seq data. Interestingly, our own analysis of day 5 somitic cells revealed multiple distinct subclusters, suggesting transcriptional heterogeneity within this population, which could have not been inferred from bulk data (*Figure 4B and C*, *Figure 4—figure supplement 4*). Importantly, neither of these papers report formation of somite-like structures, suggesting that transcriptional similarity alone is not sufficient to predict morphological features, in line with our screening results showing that average marker gene expression is not a good predictor of *in vitro* induction efficiency of somite-like structures (*Figure 3D and E*, *Figure 3—figure supplement 2A–C*).

While expression patterns of canonical somitic marker genes seem to be conserved in humans, it will be interesting to perform detailed gene expression analysis to identify putative human-specific genes of somite differentiation. Since our Somitoid system is reproducible, it could serve as a versatile platform to perform functional screens of human-specific or disease-relevant genes using CRISPR/Cas9 or small-molecule inhibitor libraries. Somitoids thus provide a powerful *in vitro* system for studying the regulation and dynamics of human somitogenesis, including somite formation.

One limitation of this work is that the current protocol does not produce an AP axis and therefore does not generate somite-like structures in a bilaterally symmetric fashion as in the vertebrate embryo. A similar phenotype was recently reported in mouse gastruloids grown under similar signaling conditions that were embedded in Matrigel to promote self-organization into trunk-like structures (TLS; *Veenvliet et al., 2020*). Chemical modulation of BMP and WNT signaling pathways in Matrigel-embedded gastruloids resulted in the formation of somite-like structures arranged like a bunch of grapes, similar to what we observed in our system. In standard culture conditions, gastruloids and TLS recapitulate the axial organization of the embryo, which is missing in our Somitoids (*Beccari et al., 2018*; *Moris et al., 2020*; *Veenvliet et al., 2020*). To expand the patterning and morphogenetic potential of our Somitoid system, our approach could be combined with a microfluidics setup to generate spatio-temporally controlled morphogen gradients (*Manfrin et al., 2019*). In summary, Somitoids provide a scalable, reproducible, and easy to manipulate platform to study molecular networks underlying the differentiation of PM, as well as the morphogenetic processes of somite formation. Furthermore, Somitoids represent a promising *in vitro* system to study congenital diseases that are linked to the human segmentation clock and somite formation, such as congenital scoliosis.

# Materials and methods

## Key resources table

| Reagent type (species) or resource | Designation | Source or reference | Identifiers | Additional information |
|---|---|---|---|---|
| Cell line (*Homo sapiens*) | NCRM1 | RUCDR Infinite Biologics | RRID:CVCL_1E71 | hiPSC line |
| Cell line (*H. sapiens*) | ACTB-GFP | Allen Institute | AICS-0016-184; RRID:CVCL_JM16 | hiPSC line |
| Cell line (*H. sapiens*) | WTC-11 | Coriell Institute | GM25256; RRID:CVCL_Y803 | hiPSC line |
| Antibody | Anti-CDH2/N-CADHERIN (rabbit polyclonal) | Abcam | ab18203 | (1:400) |
| Antibody | Anti-PAX3 (mouse monoclonal) | DSHB | Pax3-c | (1:250) |
| Antibody | Anti-SOX2 (goat polyclonal) | R&D Systems | AF2018 | (1:200) |
| Antibody | Anti-T/BRACHYURY (rabbit monoclonal) | Abcam | ab209665 | (1:200) |
| Antibody | Anti-TBX6 (rabbit polyclonal) | Abcam | ab38883 | (1:300) |
| Antibody | Anti-TCF15 (rabbit polyclonal) | Abcam | ab204045 | (1:50) |
| Antibody | Anti-MYH1 (mouse monoclonal) | DSHB | MF20-c | (1:300) |
| Sequence-based reagent | RT-qPCR primers | | | *Supplementary file 2* |
| Chemical compound, drug | CHIR99021 | Sigma | SML1046 | |
| Chemical compound, drug | LDN193189 | Stemgent | 04-0074 | |
| Chemical compound, drug | Y-27362 dihydrochloride | Tocris | 1254 | |

## hiPSC culture and 3D differentiation

hiPSCs were maintained on Matrigel-coated plates (Corning, Cat# 354277) in mTeSR1 media (Stem Cell Technologies, 85870) using maintenance procedures developed by the Allen Institute for Cell Science (https://www.coriell.org/1/AllenCellCollection). NCRM1 iPSCs were obtained from RUCDR Infinite Biologics, ACTB-GFP (cell line ID: AICS-0016 cl.184) fluorescent reporter iPSC line was obtained from the Allen Institute for Cell Science, and the WTC-11 (GM25256) cell line was obtained from the Coriell Institute for Medical Research. All cell lines were tested for mycoplasma contamination. We verified cell line identity by immunostaining for pluripotency markers POU5F1 and SOX2.

For generation of PM organoids, 500 dissociated iPSCs resuspended in mTeSR1 media containing 10 µM Y-27362 dihydrochloride (ROCKi; Tocris Bioscience, Cat# 1254) and 0.05% PVA were dispensed into 96-well U-bottom non-adherent suspension culture plates (Greiner Bio-One, 650185) and allowed to aggregate for 24 hr. To induce PM differentiation, 24-hr-old pluripotent spheroids were subjected to CL media consisting of RHB Basal media (Takara/Clontech, Cat# Y40000), 5% KSR (Thermo Fisher Scientific, Cat# 10828028) with 10 µM CHIR99021 (Sigma-Aldrich, Cat# SML1046), 0.5 µM LDN193189 (Stemgent, Cat# 04-0074), and 5 µM ROCKi for the first 24 hr. Organoids were cultured in CL media without ROCKi from 24 to 72 hr of differentiation. On day 3 (72–120 hr), CL media was supplemented with 20 ng/ml FGF2 (PeproTech, Cat# 450-33). On day 4, organoids were cultured in basal media only, without the addition of signaling factors.

## Human sclerotome and dermomyotome differentiation

To further differentiate Somitoids towards sclerotome fate, day 5 somite-stage organoids were treated with 5 nM of Shh agonist SAG 21k (Tocris, Cat# 5282) and 1 µM of Wnt inhibitor C59 for 3 days as previously described (*Loh et al., 2016*). Organoids were subsequently differentiated towards cartilage by culturing them in the presence of 20 ng/ml BMP4 (R&D Systems, Cat# 314 BP-010) for 6 days.

To differentiate Somitoids towards dermomyotome, day 5 somite-stage organoids were treated with CHIR99021 (3 µM), GDC0449 (150 nM), and BMP4 (50 ng/ml) for 48 hr as described previously (*Loh et al., 2016*; *Matsuda et al., 2020*).

### *In vitro* skeletal muscle differentiation

Day 7 organoids differentiated towards dermomyotome fate were dissociated with Accutase, resuspended in muscle induction medium containing ROCK inhibitor Y27632 and seeded (1.5–2.5 × 10$^5$ cells per well) onto Matrigel-coated 12-well plates. To induce human skeletal muscle cells, we used an N2/horse-serum-containing induction medium as previously described (*Matsuda et al., 2020*). In brief, DMEM/F12 containing GlutaMAX (Thermo Fisher Scientific, Cat# 10565018), 1% insulin-transferrin-selenium (Thermo Fisher Scientific, Cat# 41400045), 1% N-2 Supplement (Thermo Fisher Scientific, Cat# 17502-048), 0.2 penicillin/streptomycin (Sigma-Aldrich, Cat# P4333-100ML), and 2% horse serum (Sigma-Aldrich, Cat# H1270-100ML). Medium was changed every other day. Day 45 cells were fixed in 4% PFA and immunostained for myosin heavy chain (DSHB, MF20-c, 1:300).

### Small-molecule inhibitor screens

For the systematic small-molecular inhibitor screen, PM organoids were generated and differentiated until day 3 (PSM) of our protocol. On day 3, media was replaced with fresh media containing combinations of small-molecule inhibitors targeting the FGF, WNT, BMP, and TGF-β signaling pathways at indicated concentrations. For targeting the WNT pathway, we used C59 (Tocris, Cat# 5148), XAV939 (Tocris, Cat# 3748), and CHIR99021 (Sigma-Aldrich, Cat# SML1046). For inhibiting the FGF pathway, we used PD173074 (Sigma-Aldrich, Cat# P2499). For inhibiting the BMP pathway, we used LDN193189 (Stemgent, Cat# 04-0074). For inhibition of the TGF-β pathway, we used A-83-01 (Tocris, Cat# 2939). Media was changed daily. We analyzed three replicates per condition in the primary screen and five replicates per condition in the secondary screen.

### Immunostaining

For organoid whole-mount immunostaining, organoids were collected in cold PBS and fixed in 4% paraformaldehyde for 1–2 hr depending on size/stage. Organoids were washed in PBS and PBSFT (PBS, 0.2% Triton X-100, 10% FBS), and blocked in PBSFT + 3% normal donkey serum. Primary antibody incubation was performed in the blocking buffer overnight at 4°C on a rocking platform. After extensive washes in PBSFT, secondary antibody incubation (1:500, all secondary antibodies were raised in donkey) was performed overnight in PBSFT. Organoids were washed first in PBSFT and, for the final washes, were transferred to PBT (PBS, 0.2% Triton X-100, 0.2% BSA), followed by 50% glycerol in PBT and 70% glycerol in PBT prior to mounting. Hoechst (1:2000) was added to the last PBSFT wash. A list of primary antibodies is provided in Table S1.

### Confocal and time-lapse microscopy

All whole-mount immunostaining images were collected with a Nikon A1R point scanning confocal with spectral detection and resonant scanner on a Nikon Ti-E inverted microscope equipped with a Plan Apo VC ×20 objective (NA 0.75). Alexa-488, Alexa-594, Alexa-647 fluorophores coupled to secondary antibodies were excited with the 488 nm, 561 nm, and 647 nm laser lines from a Spectral Applied Research LMM-5 laser merge module with solid-state lasers (selected with an AOTF) and collected with a 405/488/561/647 quad dichroic mirror (Chroma). For time-lapse experiments, images were acquired with a Yokagawa CSU-X1 spinning disk confocal on a Nikon Ti inverted microscope equipped with a Plan Apo ×20 objective (NA 0.75) and a Hamamatsu Flash4.0 V3 sCMOS camera. Samples were grown on six-well glass-bottom multiwell plates with no. 1.5 glass (Cellvis, Cat# P06-1.5H-N) and mounted in a OkoLab 37°C, 5% $CO_2$ cage microscope incubator warmed to 37°C. Images were collected every 15 min using an exposure time of 800 ms. At each timepoint, 30 z-series optical sections were collected with a step size of 2 μm. Multiple-stage positions were collected using a Prior Proscan II motorized stage. Z-series are displayed as maximum z-projections, and gamma, brightness, and contrast were adjusted (identically for compared image sets) using Fiji/ImageJ (*Schindelin et al., 2012*; https://imagej.net/Fiji).

### Automated image segmentation and analysis

Automated image analysis, including background denoising, segmentation, and feature extraction, was done using ImageJ/Fiji macro language run in batch mode to process the entire screen dataset. First, binary masks were generated from the Hoechst (nuclear stain) channel by denoising the image (Gaussian blur, sigma = 5) followed by applying Li's Minimum Cross Entropy thresholding method (*Li*

*and Tam, 1998*) and refining binary masks through several rounds of erosion/dilation steps. Next, binary masks were converted to selections and added to the region of interest (ROI) Manager. Finally, ROIs were used to perform diameter measurements of organoids. For Pax3 measurements, Hoechst and Pax3 channels were first denoised using a Gaussian blur filter (sigma = 10) and then used to create a normalized Pax3 image by dividing the Pax3 channel with the Hoechst channel. Next, ROIs based on Hoechst binary masks were applied to the Pax3 normalized image to extract fluorescence intensity measurements for each z-slice. Finally, mean Pax3 intensity values for each organoid were calculated and compared.

## Quantification of per organoid number of somite-like structures for secondary screen

For the primary and secondary screens, images were acquired on a Nikon A1R point scanning confocal microscope. For each organoid, 66 z-series optical sections were collected with a step size of 2 μm. Quantification of somite-like structures for the secondary screen was done by blinded manual scoring, considering the following criteria:

1. Nuclear expression of somitic marker PAX3.
2. Accumulation of NCAD around a central cavity.
3. Radial arrangement of PAX3+ columnar cells around the central cavity (rosette-like structure).

## Quantification of organoid and human somite sizes

Carnegie stage 9–11 human embryonic somite data was obtained from the Virtual Human Embryo Project (https://www.ehd.org/virtual-human-embryo/). Somite sizes of human embryos were measured using the Ruler Tool on the Virtual Human Embryo website along the mediolateral and dorsoventral axis of the embryo. The slice with the largest diameter of each somite was used for measurements. Sizes of somite-like structures of day 5 organoids were measured along the X and Y axes of the image since, unlike in the embryo, they do not exhibit morphological anisotropies. Somite areas were approximated by using the two diameter measurements from each somite-like structure to calculate the area of the resulting rectangle.

## RNA extraction, reverse transcription, and qPCR

Organoids were harvested in Trizol (Life Technologies, Cat# 15596-018), followed by precipitation with chloroform and ethanol and transfer onto PureLink RNA Micro Kit columns (Thermo Fisher, Cat# 12183016) according to the manufacturer's protocol, including on-column DNase treatment. A volume of 22 μl RNase-free water was used for elution, and RNA concentration was measured with a Qubit Fluorometer. Typically, between 0.2 and 1 μg of RNA was reverse transcribed using Superscript III First Strand Synthesis kit (Life Technologies, Cat# 18080-051) and oligo-dT primers to generate cDNA libraries.

For real-time quantitative PCR, cDNA was diluted 1:30-1:50 in water and qPCR was performed using the iTaq Universal SYBR Green kit (Bio-Rad, Cat# 1725124). Each gene-specific primer and sample mix was run in triple replicates. Each 10 μl reaction contained 5 μl 2X SYBR Green Master Mix, 0.4 μl of 10 μM primer stock (1:1 mix of forward and reverse primers), and 4.6 μl of diluted cDNA. qPCR plates were run on a Roche LightCycler 480 Real-Time PCR system with the following cycling parameters: initial denaturation step (95°C for 1 min), 40 cycles of amplification and SYBR green signal detection (denaturation at 95°C for 5 s, annealing/extension and plate read at 60°C for 40 s), followed by final rounds of gradient annealing from 65 to 95°C to generate dissociation curves. Primer sequences are listed in *Supplementary file 2*, Table S2. All unpublished primers were validated by checking for specificity (single peak in melting curve) and linearity of amplification (serially diluted cDNA samples). For relative gene expression analysis, the ΔΔCt method was implemented using the R package 'pcr' (https://cran.r-project.org/web/packages/pcr/). PP1A was used as the housekeeping gene in all cases. Target gene expression is expressed as fold-change relative to undifferentiated iPSCs.

## Preparation of single-cell suspensions for scRNA-seq

Cell dissociation protocols were optimized to achieve single-cell suspensions with >90% viable cells and low number of doublets.

Organoids collected at days 1, 2, 3, and 5 of our differentiation protocol were pooled in prewarmed PBS, transferred to prewarmed Accutase, and incubated for 5–7 min at 37°C. For day 1 organoid cell suspension, 30 organoids were pooled. For day 2 organoid cell suspension, 15 organoids were pooled. For day 3 cell suspension, eight organoids were pooled. For day 5 cell suspension, five organoids were pooled. Organoids were briefly rinsed in PBS, then transferred to 500 µl PBS/0.05% BSA and carefully triturated to generate a single-cell suspension. Cell suspension was run through a cell strainer (Falcon, Cat# 352235) and transferred to a 1.5 ml tube. Cells were spun down at 250 × *g* for 3 min at 4°C. Cell pellet was resuspended in 25 µl PBS/0.05% BSA, cell concentration and viability were measured using an automated cell counter, and cell suspension was further diluted as appropriate to reach the optimal range for 10× (700–1200 cells per µl). Cells were subjected to scRNA-seq (10X Genomics, Chromium Single Cell 3' v3) aiming for the following target cell numbers: day 1, 3000 cells; day 2, 4000 cells; day 3, 5000 cells; day 5, 6000 cells. Estimated actual cell numbers collected were day 1, 2930 cells; day 2, 4977 cells; day 3, 5968 cells; day 5, 4841 cells. Single-cell libraries were generated using standard protocols. Libraries were sequenced together on a NovaSeq 6000 system resulting in 800 million reads.

## Analysis of scRNA-seq data

Statistics and plots were generated using R version 4.0.2 'Taking Off Again' and Seurat version 3.0 (*Stuart et al., 2019*).

## QC analysis/processing of scRNA-seq data

Cell Ranger pipeline (10X Genomics, version 4.0.0) was used to demultiplex the raw base call files, generate FASTQ files, perform the alignment against the human reference genome (GRCh38 1.2.0), and generate the count matrices.

For the initial QC, we determined the following thresholds for filtering out low-quality cells: UMI counts less than 500, gene counts less than 200, mitochondrial fraction above 0.2, and a complexity score of less than 0.8 (calculated as log10 (genes)/log10 (UMIs)).

## Low-dimensional embedding and clustering

After QC filtering, we normalized our dataset using the sctransform (*Hafemeister and Satija, 2019*) framework, which is part of the Seurat package. To regress out confounding variation in our dataset, we performed cell cycle scoring and determined mitochondrial mapping percentage using standard workflows. Next, we performed principal component analysis and determined the K-nearest neighbor graph using the first 40 principal components. We then applied the Leiden clustering algorithm using a parameter range from 0.1 to 1.0 to determine the best resolution/number of clusters, which reflected biological differences (FindClusters, resolution = 0.1–1.0). Clusters were visualized on a Uniform Manifold Approximation and Projection (UMAP) embedding (RunUMAP, dims = 1:40). To determine optimal resolution for clustering and assign cell types for each cluster, we visualized sets of known marker genes for each predicted cell type on UMAP plots. Prior to marker gene identification and final assignments of cluster identities, we also checked additional quality control metrics (UMI count, gene count, mitochondrial gene ratio) to exclude low-quality clusters from downstream analyses. Through iterative analysis, we determined Leiden clustering with resolution = 0.8, resulting in 22 clusters, to best capture biological variation of the dataset. Using a combination of quality control metrics and unbiased marker gene identification for each cluster (see below), we excluded seven smaller low-quality clusters (as determined by QC metrics and/or expression of stress signature genes) from further downstream analysis (15 clusters after filtering).

## Identification of differentially expressed genes

Marker genes for every cluster were identified by a two-sided Wilcoxon rank-sum test comparing cells from each cluster to all other cells in the combined dataset. Genes were considered differentially expressed if the log2 fold-change average expression in the cluster is equal to or greater than 0.25 relative to the average expression in all other clusters combined, and the adjusted p-value<0.05. Multiple comparison correction was performed using the Bonferroni method. Identified marker genes for the top 20 differentially expressed transcripts are listed in *Figure 4—figure supplement 4A*. The

full list of differentially expressed genes, ranked by adjusted p-values and associated fold-changes, is provided in *Supplementary file 1*, Table S1.

## Acknowledgements

We would like to acknowledge support of the Nikon Imaging Center at Harvard Medical School for image acquisition and consulting.

## Additional information

### Funding

| Funder | Grant reference number | Author |
|---|---|---|
| National Institutes of Health | P01GM099117 | Christoph Budjan Sahand Hormoz |
| National Heart, Lung, and Blood Institute | R01HL158269 | Christoph Budjan Sahand Hormoz |
| Chan Zuckerberg Initiative | 2018-183143 | Christoph Budjan Sahand Hormoz |

The funders had no role in study design, data collection and interpretation, or the decision to submit the work for publication.

### Author contributions

Christoph Budjan, designed, optimized and performed the experiments; analyzed the experimental and single-cell RNA-sequencing data, Conceptualization, Formal analysis, Investigation, Methodology, Writing – original draft, Writing – review and editing; Shichen Liu, Methodology, assisted with single-cell sample collection and sequencing library preparation; Adrian Ranga, Methodology; Senjuti Gayen, Formal analysis; Olivier Pourquié, Funding acquisition, Supervision; Sahand Hormoz, Funding acquisition, Supervision, Writing – original draft, Writing – review and editing

### Author ORCIDs

Christoph Budjan (iD) http://orcid.org/0000-0002-1563-9750
Shichen Liu (iD) http://orcid.org/0000-0002-0964-6559
Adrian Ranga (iD) http://orcid.org/0000-0002-6400-9472
Sahand Hormoz (iD) http://orcid.org/0000-0002-4384-4428

### Decision letter and Author response

Decision letter https://doi.org/10.7554/eLife.68925.sa1
Author response https://doi.org/10.7554/eLife.68925.sa2

## Additional files

### Supplementary files

• Supplementary file 1. Supplemental Table 1: cluster-based marker genes of single-cell RNA-sequencing (scRNA-seq) dataset.
• Supplementary file 2. Supplemental Table 2: RT-qPCR primer sequences.
• Transparent reporting form

### Data availability

Sequencing data has been deposited in GEO under accession code GSE194214; All data used to generate the figures are included in the manuscript as Source Data files.

The following dataset was generated:

| Author(s) | Year | Dataset title | Dataset URL | Database and Identifier |
|---|---|---|---|---|
| Christoph B, Shichen L, Sahand H | 2021 | Paraxial mesoderm organoids model development of human somites | https://www.ncbi.nlm.nih.gov/geo/query/acc.cgi?acc=GSE194214 | NCBI Gene Expression Omnibus, GSE194214 |

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
