## [Editor Report]

Budjan et al. describe an organoid protocol to obtain somite-like structures from human iPSCs. Using defined culture media, the authors describe the formation after 5 days *in vitro* of organoids that express a variety of PSM differentiation markers, such as the segmentation clock gene Hes7 and Pax3, thus recapitulating the time course of expression markers typically observed along PSM and somite early differentiation.

---

## [Decision Letter]

**Decision letter after peer review:**

Thank you for submitting your article "Paraxial mesoderm organoids model development of human somites" for consideration by *eLife*. Your article has been reviewed by 2 peer reviewers, and the evaluation has been overseen by Marianne Bronner as the Senior and Reviewing Editor. The reviewers have opted to remain anonymous.

The reviewers have discussed their reviews with one another, and the Reviewing Editor has drafted the letter below. As you will see, the reviewers raise some significant concerns, particularly regarding the morphological resemblance of somitoids to somites in vivo. If you feel you can address these concerns, we would be open to receiving a significantly revised version for full in depth review. Equally, we realize that you may prefer to go to a different journal at this juncture rather than make extensive revisions.

Essential revisions:

One of the great utilities of organoids is their self-organising ability which enables them to adopt morphological and molecular features that closely resemble their in vivo counterparts. While we appreciate that challenges of utilizing human cell lines, a major concern with the current work it that the human somitoids do not morphologically resemble somites. To remedy this:

1. In order to distinguish these from differentiating mesoderm, the authors need to provide solid morphological evidence that somitoids can generate somite-like structures. This would include demonstrating the presence of pseudo-stratified, bottle shaped epithelial cells, with apical/basal inside/out polarity.

The full reviews are included to help with a possible revision.

*Reviewer #1 (Recommendations for the authors):*

1) Three human iPSC lines are mentioned in the methods and replicates are mentioned throughout the text. The authors should make clear for each data set/figure presented which iPSC line(s) was used. Moreover, they should comment any between-line variability in the differentiation protocol. I can only find inter-organoid (so within-experiment) variability in the supplemental data. The extent of technical (between experiment) and biological variation is not clear.

2) It is not indicated clearly anywhere exactly what the authors are scoring as a somite. The clearest depiction is Figure 5A where rosettes of TCF15 with apical F actin can be observed. Is each of these scored as a somite? A higher magnification should be provided in Figure 3 showing clearly how somites are scored.

3) The combinatorial FGFRi/Wnti treatments using various small molecules mentioned in Figure 2C, D, 3B, C, D does not impact the overall understanding of the differentiation protocol. Moreover, data suggests that inhibition of FGF and Wnt signalling using small molecules is not required to derive PAX3+ somites. Just extending the CL+ FGF by 24hrs (day3 to day4) and then changing media to basal condition (no growth factors) for 24hrs is sufficient for differentiation. This condition also results in significantly higher PAX3+ cells and somite number (Figure 3C, D). Why was this condition not utilised further? Why is the use of FGFRi/Wnti essential?

4) d5 Somites are TCF15+ (Figure 4E, 5A) how different are these cells from the Somatic mesoderm population published in Diaz-Cuadros et al., 2020; Matsuda et al., 2020?

5) Figure1A and 2A: What is the rationale behind choosing D3 for modulating the FGFi/Wnti treatments. D4 aPSM (Figure 1A) are further committed towards somitogenesis and wouldn't modulating small molecule treatment provide a better output?

6) Figure 2C and D: 500 cells was chosen for all downstream experiments based on f-actin arrangement (line111). Authors need to significantly elaborate on this rationale. Does this relate to the definition of an in vitro somite? The data seem to suggest that using 500 cells (in 24hour condition) generates smaller organoids with no significant difference in PAX3 expression intensity.

7) Figure 4: Which differentiation condition was eventually used for sc-RNA seq analysis (Figure 4). Was it the CL+FGF condition or the FGFi/Wnti condition? This information is absolutely critical to determine the ultimate conclusion from Figure 1-3.

8) Figure 5: For sclerotome differentiation and other comparisons in this figure CL+FGF condition has been used (with no use of FGFi/Wnti for somite formation). With significant focus given on timed use of FGFi/Wnti for acquiring somite fate in Figure 1,2,3 why has the alternate method been used in this section?

9) Figure 5B and line 202, 2019-210: The in vitro organoid size is significantly larger than in vivo human stage9-10. Why has this difference been overlooked to claim "our organoid-derived somites share spatial and molecular features as well as overall size with their in vivo counterparts".

*Reviewer #2 (Recommendations for the authors):*

It seems therefore that despite the evident amount of work already placed in this study, this is rather incremental addition to existing knowledge rather than a breakthrough in the field.

---

## [Author Response]

Essential revisions:One of the great utilities of organoids is their self-organising ability which enables them to adopt morphological and molecular features that closely resemble their in vivo counterparts. While we appreciate that challenges of utilizing human cell lines, a major concern with the current work it that the human somitoids do not morphologically resemble somites. To remedy this:1. In order to distinguish these from differentiating mesoderm, the authors need to provide solid morphological evidence that somitoids can generate somite-like structures. This would include demonstrating the presence of pseudo-stratified, bottle shaped epithelial cells, with apical/basal inside/out polarity.

We thank the reviewers and editor for the feedback on our manuscript. We agree that it is important to clearly show the morphological features of our *in vitro* generated somites because it is one the main advances of our 3D Somitoid system over previously published 2D models. In the revised manuscript, we have included high-magnification images of somite-like structures, which clearly show the shape of the somitic cells and the polarity of these epithelial cells by staining for multiple polarity markers (new Figure 3-Supplemental Figure 1; new Figure 3-Video 1). Consistent with the morphology and polarity of *in vivo* somites, we observe PAX3+/TCF15+ bottle-shaped cells radially arranged around a central cavity, which form rosette-like structures that are approximately the same size as the Carnegie stage 11 *in vivo* somites (Figure 5B). This can be observed in multiple new images added to our revised manuscript (Figure 3-Supplemental Figure 1A-A’’, Figure 3-Supplemental Figure 1B-B’’, Figure 2-Supplemental Figure 3A,B). Additionally, apical surface markers F-ACTIN (Figure 5A) and N-CADHERIN (Figure 3-Supplemental Figure 1) are both expressed around the central cavity, suggesting that the apical side of the somitic cells is facing the inside of the somite structure, again consistent with their in vivo counterparts. This is particularly evident in the newly added high-magnification images (Figure 3-Supplemental Figure 1) and accompanying video showing a full confocal z-stack through the *in vitro* somites (Figure 3- Video 1). Together with our protein (Figure 3B,C and Figure 5A) and gene expression data (both in bulk (Figure 1C, Figure 5C, Figure 1-Supplemental Figure 1B) and at the single-cell level (Figure 4, Figure 4-Supplemental Figure 1-5)), and directed differentiation experiments of Somitoid-derived cells towards sclerotome (Figure 5C) and dermomyotome (newly added Figure 5-Supplemental Figure 1), we conclude that our *in vitro* somites are molecularly, morphologically, and functionally equivalent to *in vivo* somites.

The full reviews are included to help with a possible revision.Reviewer #1 (Recommendations for the authors):1) Three human iPSC lines are mentioned in the methods and replicates are mentioned throughout the text. The authors should make clear for each data set/figure presented which iPSC line(s) was used. Moreover, they should comment any between-line variability in the differentiation protocol. I can only find inter-organoid (so within-experiment) variability in the supplemental data. The extent of technical (between experiment) and biological variation is not clear.

The reviewer brings up a very important point. To clarify which cell lines were used for each figure, we have added the information to each figure caption in the revised manuscript. We have also specified the number of organoids used to quantify variation for each experiment in the figure captions. As described below, we have conducted new experiments to further quantify the technical variability across experiments, as well as quantify the efficacy of our optimized differentiation protocol when applied to genetically independent cell lines.

For our initial screens (Figures 2 and 3), we only used one cell line (NCRM1 hiPSCs). As mentioned by the reviewer, we measured variability across individual organoids for each condition of the screen to help identify the conditions that minimized variability and produced the most reproducible organoids (Figure 2-Supplemental Figure 2B; Figure 3-Supplemental Figure 1B).

To analyze technical variability of our optimized protocol, we have repeated our optimized protocol two more times and quantified the number of somites across 10 organoids for each experiment (see newly added Figure 3-Supplemental Figure 3). Inter-organoid variability was comparable to our initial results from the secondary screen (CV for Experiment 1 = 17% and CV for Experiment 2 = 9.8%; see also Figure 3-Supplemental Figure 2B). Furthermore, mean and median are very comparable between the two additional experiments (Experiment 1: 39+/-8 (mean+/-std), median=40; Experiment 2: 43+/-4 (mean+/-std), median=41, p-val = 0.16). Please note that the absolute number of somites in our new experiments has increased compared to our initial screen (see Figure 3). This is a result of improvements in both our immunostaining protocol as well as the image acquisition workflow. The two new experiments both used the same improved staining and image acquisition workflow and are therefore comparable with each other.

To extend our variability analysis to other cell lines, we tested the following cell lines: our original cell line (NCRM1), the WTC cell line released by the Conklin Laboratory at the Gladstone Institute, and a reporter cell line, ACTB-GFP, from the Allen Cell Collection. We tested our optimized protocol alongside another high scoring condition in all three cell lines:

– CL+FGF2 for 24h, basal media for 24h (our optimized protocol)

– WNTi^hi^(C59, 2 µM) for 48h

Applying our optimized protocol to two other cell lines confirmed that our protocol is reproducible across different genetic backgrounds/cell lines. NCRM1, average somite number = 43+/-4 (mean+/-std); ACTB-GFP, average somite number = 40+/-6; WTC, average somite number = 33+/-4. We also tested one additional top scoring condition (C59, 2µM for 48 hours) in all three cell lines. Notably, for this condition, the ACTB-GFP derived Somitoids (32+/-4 (mean+/-std)) showed a higher average number of somites compared with the Somitoids derived from the other cell lines (NCRM1, 20+/-3; WTC, 17+/-4). However, our optimized protocol resulted in the highest number of somites across all cell lines.

2) It is not indicated clearly anywhere exactly what the authors are scoring as a somite. The clearest depiction is Figure 5A where rosettes of TCF15 with apical F actin can be observed. Is each of these scored as a somite? A higher magnification should be provided in Figure 3 showing clearly how somites are scored.

We thank the reviewer for an insightful comment and question. Following the suggestion of the reviewer, we generated new organoids with somite structures and imaged them at a higher magnification (shown in the newly added Figure 3-Supplement Figure 1). The higher magnification clearly shows the ‘bottle-shaped’ morphology of the cells that comprise the somites in that the apical surface of these columnar cells is typically smaller than their basal side. These higher-magnification images also more clearly show the rosette structure (radial arrangement of columnar cells) with a central cavity in which NCAD is highly expressed. Nuclear expression of PAX3 is also clearly visible in these cells. These empirical observations were the criteria used to manually identify somites in the images acquired from the organoid screen. We have clearly described these scoring criteria in a new paragraph added to the Methods section (section titled “Quantification of per organoid somite numbers for secondary screen”).

To further illustrate the scoring criteria for identifying somites, we have also included a new figure panel, Figure 3B. In this figure, we show examples of somite quantification using our scoring criteria. Two z-sections are used as examples and the manually scored somites are highlighted using dashed circles.

Additionally, we generated videos of z-stacks acquired on a confocal microscope showing examples of organoids with and without somite-like structures based on our scoring criteria using PAX3 and NCAD staining (Newly added Figure 3-Video 1).

3) The combinatorial FGFRi/Wnti treatments using various small molecules mentioned in Figure 2C, D, 3B, C, D does not impact the overall understanding of the differentiation protocol. Moreover, data suggests that inhibition of FGF and Wnt signalling using small molecules is not required to derive PAX3+ somites. Just extending the CL+ FGF by 24hrs (day3 to day4) and then changing media to basal condition (no growth factors) for 24hrs is sufficient for differentiation. This condition also results in significantly higher PAX3+ cells and somite number (Figure 3C, D). Why was this condition not utilised further? Why is the use of FGFRi/Wnti essential?

We apologize to the reviewer if this important point was not clear in the main text. We indeed end up using CL+FGF2 from Day 3 to Day 4 followed by culture in basal media as our preferred differentiation protocol for all subsequent experiments (shown in Figure 4-5), including our single-cell RNA-seq experiment. This optimized protocol is a direct conclusion from our systematic screens and was not obvious prior to performing these screens. We have revised the manuscript so that this point is clear to the reader (see line numbers 176-183 on page 6).

Our screens focused on modulating the timing and duration of WNT and FGF signaling pathways (in addition to BMP and TGF-β) based on long-established *in vivo* evidence (Aulehla et al., 2008, 2003; Delfini et al., 2005; Dubrulle et al., 2001; Dunty et al., 2008; Greco et al., 1996; Yamaguchi et al., 1999)(Loh et al., 2016; Xi et al., 2017)(Aulehla et al., 2008, 2003; Delfini et al., 2005; Dubrulle et al., 2001; Dunty et al., 2008; Greco et al., 1996; Yamaguchi et al., 1999) and recently published *in vitro* differentiation protocols (Chal et al., 2015; Loh et al., 2016; Matsuda et al., 2020; Sakurai et al., 2012; Tan et al., 2013). To make the rationale for designing the screens more clear, we describe it in the revised manuscript and include citations for *in vivo* evidence and *in vitro* differentiation protocols using hPSCs (see line numbers 97-105 on page 4).

Importantly, one main conclusion from the screen and our in-depth single-cell RNA-seq analysis is that while it is not necessary to actively inhibit WNT and/or FGF signaling to induce somite fate *in vitro*, day 5 somite-like cells from our optimized protocol autonomously downregulate WNT target genes (DKK1, AXIN2, WNT3A, RSPO3; Figure 4-Supplemental Figure 2) and FGF target genes (FGF8, FGF17, SPRY4, DUSP6/MKP3, SEF/IL17RD) in addition to NOTCH target genes (DLL1, DLL3, HES7). To illustrate this, we have generated a heatmap of single-cell gene expression of canonical target genes for these signaling pathways (see newly added Figure 4-Supplemental Figure 5) over the course of the differentiation protocol.

4) d5 Somites are TCF15+ (Figure 4E, 5A) how different are these cells from the Somatic mesoderm population published in Diaz-Cuadros et al., 2020; Matsuda et al., 2020?

The reviewer raises a valid and important point. To address the reviewer’s question, we compared the single-cell RNA-seq data from Diaz-Cuadros et al., with our own single-cell RNA-seq data. The 2D differentiated cells in Diaz-Cuadros et al., at the final time point of their experiment do not show a clear somitic cell state signature. PAX3, MEOX1, MEOX2, FOXC2, UNCX, and TBX18 are not expressed (compared for example to FOXC1). FOXC1 does not appear to be a somite-specific marker as it is expressed in early and late PSM-like cells as seen in our own data, starting on day 2 (Figure 4-Supplemental Figure 3). In the Diaz-Cuadros et al., dataset, TCF15 is not expressed uniformly in all cells nor specifically in the late-stage cells (see UMAP plots and single-cell heatmap in the newly generated figure). Conversely, TCF15 is specifically expressed in day 5 somitic cells in our dataset both in a uniform manner and at high levels (as shown in Author response image 1 and Figure 4C).

**Author response image 1. sa2fig1:** 

We also analyzed the bulk RNA-seq data in Matsuda et al., 2020 as shown in Figure 1B of their paper. They show the expression of 4 somite markers (TCF15, MEOX1, PAX3, and RIPPLY1). As can be seen in Author response image 1 (and Figure 4-Supplemental Figure 3), all markers highlighted in their paper are specifically expressed in our day 5 somitic cell population, including RIPPLY1 (see also updated Figure 4-Supplemental Figure 3). Beyond comparison of marker gene expression, it is difficult to assess the similarities and differences with the Matsuda dataset since their data lacks single-cell resolution. Thus, heterogeneity and efficiency of somitic fate induction within their cell population is unclear.

Finally, neither of these two papers report the formation of somite-like structures. To make this comparison more clear, we have added it to the Discussion section of the revised manuscript (see line numbers 291-304 on page 10).

5) Figure1A and 2A: What is the rationale behind choosing D3 for modulating the FGFi/Wnti treatments. D4 aPSM (Figure 1A) are further committed towards somitogenesis and wouldn't modulating small molecule treatment provide a better output?

This is a valid point by the reviewer which we considered when designing the screen. Based on the 2D data from Diaz-Cuadros et al., D3 and D4 cells have similar gene expression profiles. Their own analysis of clustering the single-cell gene expression profile of these two populations of cells does not distinguish D3 cells from D4 cells. This is not surprising because D3 and D4 cells are kept in the same media containing FGF and WNT agonists. *In vivo*, down-regulation of FGF and WNT signaling is correlated with differentiation to somite fate. In addition, in their data both D3 and D4 cells show HES7 oscillations, indicating that D4 cells are not significantly more differentiated than D3 cells. Finally, the comparison in Diaz-Cuadros et al., of the gene expression profiles of D3 and D4 cells to E9.5 mouse *in vivo* data shows that both populations closely resemble the posterior PSM and that D4 cells are no closer to an anterior PSM fate than D3 cells. Taken together, we chose to start our differentiation protocol with D3 cells, which also reduces the duration of the protocol by one day. Our earlier version of Figure 1 was therefore misleading and we have changed the labels on Figure 1A to ‘PSM’ to not falsely suggest that cells progress to a more anterior PSM fate under these conditions.

We have also considered whether D2 organoids could already be efficiently differentiated towards somite fate. Although D2 organoids expressed PSM markers, such as TBX6 and MSGN1 (Figure 1B), the expression was not as uniform as that observed in D3 organoids. This observation is consistent with our single-cell RNA-seq data. Although both D2 and D3 organoids express PSM makers, D2 organoids also still express primitive streak markers at higher levels such as T/BRA, MIXL1, POU5F1. Additionally, WNT target genes such as WNT3A and DKK1, which are markers of posterior PSM cells, are more highly expressed in D2 organoids compared with D3 organoids (Figure 4C, Figure 4-Supplemental Figure 2). In conclusion, based on both previously published data and our own single-cell RNA-seq and qPCR data, we concluded that day 3 of our differentiation protocol constitutes the best time window to optimize somite differentiation in our organoid system.

6) Figure 2C and D: 500 cells was chosen for all downstream experiments based on f-actin arrangement (line111). Authors need to significantly elaborate on this rationale. Does this relate to the definition of an in vitro somite? The data seem to suggest that using 500 cells (in 24hour condition) generates smaller organoids with no significant difference in PAX3 expression intensity.

We thank the reviewer for raising an important point. One of the main goals of our primary screen was to investigate whether initial cell number has any effect on the ability of organoids to robustly form somites without the generation of undesired (i.e. non-PSM derived) cell types. We concluded that organoids made from 500 cells were preferable over organoids made from 1000 cells based on the following observations:

1. We used F-ACTIN expression as a marker of somite-structure in the primary screen comparing organoids made from 500 cells with organoids made from 1000 cells. Qualitative assessment of F-ACTIN staining showed more consistent somite structures (radial arrangements of PAX3+ columnar cells with high expression of F-ACTIN in the central cavity) in organoids made from 500 cells. To make this more clear to the reader, we have included representative images showing differences of F-ACTIN expression across organoids made from 500 cells and 1000 cells in the newly added Figure 2-Supplemental Figure 3.

2. Although average PAX3 intensity is similar between organoids made from 500 cells and organoids made from 1000 cells, a higher fraction of PAX3-negative cells were observed in the organoids made from 1000 cells. We only made this assessment using qualitative observation by eye because single-cell segmentation of our images was not possible. To do so, additional nuclear or membrane markers would be required. Figure 2-Supplemental Figure 2A shows examples of organoids generated from 1000 cells which show patches of PAX3-negative cells. By contrast, organoids generated from 500 cells and treated with the same small-molecule inhibitors show more uniform PAX3 induction across the organoids.

3. Finally, we observed *sox2*^+^/PAX3- cells in organoids made from 1000 cells but did not observe these contaminating cells in organoids made from 500 cells. Figure 2-Supplemental Figure 2A shows examples of organoids from 1000 cells with patches of PAX3-negative cells. These cells are positive for *SOX2*, suggesting neural contaminants in organoids generated from 1000 cells. Organoids generated from 500 cells using the same treatment conditions, do not show *sox2*^+^ patches of cells.

Taken together, we concluded that organoids made from 500 cells were a better choice than organoids made from 1000 cells. To make this clear in the revised manuscript, we carefully outline our reasoning on page 4-5 (line numbers 118-129).

7) Figure 4: Which differentiation condition was eventually used for sc-RNA seq analysis (Figure 4). Was it the CL+FGF condition or the FGFi/Wnti condition? This information is absolutely critical to determine the ultimate conclusion from Figure 1-3.

We thank the reviewer for pointing out that the original version of the manuscript did not explicitly state which protocol was used for our scRNA-seq experiment. In the revised manuscript, we have added an explicit statement saying which protocol we employed for our single-cell RNA-seq analysis. Based on the number of somites generated, the low variation of somite numbers between organoids and across experiments, as well as high average PAX3+ levels, we chose the CL+FGF2 and basal media condition for all subsequent experiments following the screen, including the single-cell RNA-seq analysis (Figure 4) and the sclerotome/dermomyotome differentiation protocols (Figure 5 and Figure 5-Supplemental Figure 1).

8) Figure 5: For sclerotome differentiation and other comparisons in this figure CL+FGF condition has been used (with no use of FGFi/Wnti for somite formation). With significant focus given on timed use of FGFi/Wnti for acquiring somite fate in Figure 1,2,3 why has the alternate method been used in this section?

Please refer to the previous response on why the CL+FGF/basal condition was used for sclerotome differentiation.

9) Figure 5B and line 202, 2019-210: The in vitro organoid size is significantly larger than in vivo human stage9-10. Why has this difference been overlooked to claim "our organoid-derived somites share spatial and molecular features as well as overall size with their in vivo counterparts".

In the manuscript, we compare the *in vitro* somite size to its *in vivo* counterpart in the Carnegie collection. The reviewer is absolutely correct that we do observe differences in the size of the *in vitro* generated somites and their stage 9-10 *in vivo* counterparts (as quantified in Figure 5B). To further expand our *in vivo* comparison, we quantified somite size in the later stages of the Carnegie collection up to stage 11. We were not able to quantify somite size beyond stage 11 because later stages already showed further differentiation of somites into sclerotome and dermomyotome and therefore did not have a well-defined rosette structure to quantify. Comparing our *in vitro* generated somites to stage 11 human somites, we observed that they are very similar in size (see updated Figure 5B). It is therefore conceivable that our somites correspond to later stages of the Carnegie collection, at least based on their relative sizes. We have updated Figure 5 and the corresponding description of the data in the main text to reflect the updated analysis.

Reviewer #2 (Recommendations for the authors):It seems therefore that despite the evident amount of work already placed in this study, this is rather incremental addition to existing knowledge rather than a breakthrough in the field.

We are glad the reviewer found our study interesting and a step towards formation of complex structures *in vitro*. We believe that the reviewer has misunderstood the structure of the somites that we observe *in vitro*. In the revised manuscript, we have included high magnification images that more clearly show the shape of the cells and the locations at which the epithelial polarity markers are expressed (new Figure 3-Supplemental Figure 1; new Figure 3-Video 1). Consistent with *in vivo* somites, we do observe bottle-shaped cells radially arranged around a central cavity forming rosette-like structures that are the same size as their *in vivo* counterparts (Figure 5B). This can be observed in multiple images shown in our manuscript (Figure 3-Supplemental Figure 1A-A’’, Figure 3-Supplemental Figure 1B-B’’, Figure 2-Supplemental Figure 3A,B). Importantly, both F-ACTIN (Figure 5A) and N-CADHERIN (Figure 3-Supplemental Figure 1) are indeed expressed around the central cavity, suggesting that the apical side of the PAX3+/TCF15+ somitic cells is facing the inside of the somite structures. This is especially evident in the newly added high-magnification images (Figure 3-Supplemental Figure 1) and accompanying video showing a full confocal z-stack through the *in vitro* somites (Figure 3-Video 1). Taken together with our protein (Figure 3B,C and Figure 5A) and gene expression data (both in bulk (Figure 1C, Figure 5C, Figure 1-Supplemental Figure 1B)) and at the single-cell level (Figure 4, Figure 4-Supplemental Figure 1-5) and our directed differentiation experiments of our Somitoid cells to dermomyotome (Figure 5-Supplemental Figure 1) and sclerotome fates (Figure 5C), we believe that the somite structures generated by our optimized *in vitro* protocol are indeed equivalent to their *in vivo* counterparts.

References

Aulehla A, Wehrle C, Brand-Saberi B, Kemler R, Gossler A, Kanzler B, Herrmann BG. 2003. Wnt3a plays a major role in the segmentation clock controlling somitogenesis. *Dev Cell* 4:395–406. doi:10.1016/S1534-5807(03)00055-8

Aulehla A, Wiegraebe W, Baubet V, Wahl MB, Deng C, Taketo M, Lewandoski M, Pourquié O. 2008. A β-catenin gradient links the clock and wavefront systems in mouse embryo segmentation. *Nat Cell Biol* 10:186–193. doi:10.1038/ncb1679

Chal J, Oginuma M, Al Tanoury Z, Gobert B, Sumara O, Hick A, Bousson F, Zidouni Y, Mursch C, Moncuquet P, Tassy O, Vincent S, Miyanari A, Bera A, Garnier J-M, Guevara G, Hestin M, Kennedy L, Hayashi S, Drayton B, Cherrier T, Gayraud-Morel B, Gussoni E, Relaix F, Tajbakhsh S, Pourquié O. 2015. Differentiation of pluripotent stem cells to muscle fiber to model Duchenne muscular dystrophy. *Nat Biotechnol* 33:962–969. doi:10.1038/nbt.3297

Delfini M-C, Dubrulle J, Malapert P, Chal J, Pourquié O. 2005. Control of the segmentation process by graded MAPK/ERK activation in the chick embryo. *Proceedings of the National Academy of Sciences* 102:11343–11348. doi:10.1073/pnas.0502933102

Dubrulle J, McGrew MJ, Pourquié O. 2001. FGF Signaling Controls Somite Boundary Position and Regulates Segmentation Clock Control of Spatiotemporal Hox Gene Activation. *Cell* 106:219–232. doi:10.1016/S0092-8674(01)00437-8

Dunty WC Jr, Biris KK, Chalamalasetty RB, Taketo MM, Lewandoski M, Yamaguchi TP. 2008. Wnt3a/β-catenin signaling controls posterior body development by coordinating mesoderm formation and segmentation. *Development* 135:85–94. doi:10.1242/dev.009266

Greco TL, Takada S, Newhouse MM, McMahon JA, McMahon AP, Camper SA. 1996. Analysis of the vestigial tail mutation demonstrates that Wnt-3a gene dosage regulates mouse axial development. *Genes Dev* 10:313–324. doi:10.1101/gad.10.3.313

Loh KM, Chen A, Koh PW, Deng TZ, Sinha R, Tsai JM, Barkal AA, Shen KY, Jain R, Morganti RM, Shyh-Chang N, Fernhoff NB, George BM, Wernig G, Salomon REA, Chen Z, Vogel H, Epstein JA, Kundaje A, Talbot WS, Beachy PA, Ang LT, Weissman IL. 2016. Mapping the Pairwise Choices Leading from Pluripotency to Human Bone, Heart, and Other Mesoderm Cell Types. *Cell* 166:451–467. doi:10.1016/j.cell.2016.06.011

Matsuda M, Yamanaka Y, Uemura M, Osawa M, Saito MK, Nagahashi A, Nishio M, Guo L, Ikegawa S, Sakurai S, Kihara S, Maurissen TL, Nakamura M, Matsumoto T, Yoshitomi H, Ikeya M, Kawakami N, Yamamoto T, Woltjen K, Ebisuya M, Toguchida J, Alev C. 2020. Recapitulating the human segmentation clock with pluripotent stem cells. *Nature* 580:124–129. doi:10.1038/s41586-020-2144-9

Sakurai H, Sakaguchi Y, Shoji E, Nishino T, Maki I, Sakai H, Hanaoka K, Kakizuka A, Sehara-Fujisawa A. 2012. in vitro modeling of paraxial mesodermal progenitors derived from induced pluripotent stem cells. *PLoS One* 7:e47078. doi:10.1371/journal.pone.0047078

Tan JY, Sriram G, Rufaihah AJ, Neoh KG, Cao T. 2013. Efficient derivation of lateral plate and paraxial mesoderm subtypes from human embryonic stem cells through GSKi-mediated differentiation. *Stem Cells Dev* 22:1893–1906.

Xi H, Fujiwara W, Gonzalez K, Jan M, Liebscher S, Van Handel B, Schenke-Layland K, Pyle AD. 2017. in vivo Human Somitogenesis Guides Somite Development from hPSCs. *CellReports* 18:1573–1585.

Yamaguchi TP, Takada S, Yoshikawa Y, Wu N, McMahon AP. 1999. T (Brachyury) is a direct target of Wnt3a during paraxial mesoderm specification. *Genes & Development*. doi:10.1101/gad.13.24.3185